# Evaluation of Red Wine Acidification Using an E-Nose System with Venturi Tool Sampling

**DOI:** 10.3390/s23062878

**Published:** 2023-03-07

**Authors:** Esmeralda Hernández, José Pelegrí-Sebastiá, Tomás Sogorb, José Chilo

**Affiliations:** 1IGIC Institute, Campus Gandia, Universitat Politècnica de València, 46730 Gandia, Spain; 2Department of Electrical Engineering, Mathematics and Science, University of Gävle, 801 76 Gävle, Sweden

**Keywords:** acetic acid, data analysis, MOS gas sensor, Venturi effect, red wine

## Abstract

The quality of wine is checked both during the production process and upon consumption. Therefore, manual wine-tasting work is still valuable. Due to the nature of wine, many volatile components are released, and it is therefore difficult to determine which elements need to be controlled. Acetic acid is one of the substances found in wine and is a crucial substance for wine quality. Gas sensor systems may be a potential alternative for manual wine tasting. In this work, we have developed a TGS2620 gas sensor module to analyze acetic acid levels in red wine. The gas sensor module was refined according to the Venturi effect along with signal slope analysis, providing promising results. The example included in this paper demonstrates that there is a direct relationship between the slope of the MOS gas sensor response and the acetic acid concentration. This relationship is useful to evaluate the ethanol oxidation in acetic acid in red wine during its production process.

## 1. Introduction

Wine quality is an essential requirement for both producers and consumers. The quality assurance system for wine production includes the control of a number of factors in the cultivation phase and during wine production, as well as a final inspection for acceptance of the wine. The final inspection, which is still performed manually, includes a sensory evaluation and control of the technical analysis parameters’ limits.

In recent years, various technologies have been implemented with the aim of controlling wine quality; one of them is the electronic nose (E-nose), a non-invasive technology that mimics the human nose [1,2]. In studies, aromas of wine were analyzed to identify the most important volatile compounds. E-nose technology has been applied in many other areas, above all in beverage and food products. The authors of [3] used E-nose technology to diagnose biotic stress in Khasi Mandarin orange plants using specially developed sensors. The work in [4] presents a fully portable E-nose equipped with QCM sensors and online analysis tools. This E-nose can perform on-site gas measurement tasks, which is helpful not only for quick measurement tasks but also for long-term monitoring applications.

There are four important parts included in an E-nose system: the sensor set, control system, extraction method, and recognition pattern technology [5]. The E-nose system is the optimal solution for the classification and discrimination of volatile compounds present in wine, including acetic acid. Acetic acid is one of the most critical substances in wine, because it is an indicator of the wine’s acidity. The authors of [6] studied the compounds present in wine acidification during the malolactic fermentation process. In addition, several studies using E-noses can be found in the literature associated with the analysis of vinegar taste or other aromas in wine. Similarly, the authors in [7] developed an E-nose consisting of 16 thin film sensors, where MatLab was used to extract data and to program feature extraction and pattern recognition techniques. They were able to determine the aromatic compounds and aroma of wine samples. One of these chemical compounds was acetic acid. In reference [8], multisensory arrays, combined with pattern recognition techniques, were used to identify and classify the chemical compounds or aromas added to wine samples.

It is worth mentioning that there are other expensive and complex techniques used to analyze the components present in wine. Article [9] describes the development of an instrument based on NIR spectroscopy, which measures the ripening parameters of the grapes, i.e., sugars (Brix), pH, and anthocyanin concentration. Similarly, the work in [10] outlines a method to estimate the concentration of the most important phenolic compounds present in red wines: polyphenols, anthocyanins, and tannins. The instrument is based on a spectrometer.

In this work, we develop a TGS2620 gas sensor module (E-nose) with an improved measurement method based on the Venturi effect [11,12]. The purpose is to determine acetic acid levels in red wine, which indicate the quality of the wine. By analyzing the sensor’s acquisition curve, and the initial upward and final downward slopes, we can obtain more information about the acidity of the wine.

This paper is organized as follows. Section 2 presents the theory, methods, and materials for the development of the TGS2620 gas sensor module. Section 3 presents the experimental measurements and analyzes the results. Finally, Section 4 provides the conclusions of the work.

## 2. Materials and Methods

Wine is produced through the fermentation process, which converts sugars (mainly glucose and fructose) from grape juice to alcohol. In the fermentation process, many biochemical and chemical processes take place. There are several compounds that are produced as alcohols, esters, glycerol, and acids, among others [13]. Acetic acid is one of the substances found in wine, and it is a crucial substance for wine quality. Figure 1 (left) shows the flowchart of the wine process, where the red box indicates the phase involving wine-tasting control.

The fermentation process includes several phases during which many components are formed. First, the yeast adapts to the new environmental conditions, and in the next phase, the yeast population increases, which leads to the consumption of some nutrients. Then, when the nutrient levels are low, the yeast starts to die and the formation of ethanol (Figure 1 right (a)) and other substances begins. The yeast will consume all the sugars during respiration and fermentation phases, which results in a glycolysis process. During the glycolysis process, each molecule of hexose generates two molecules of pyruvate, four molecules of adenosine triphosphate, and one molecule of reduced nicotinamide adenine dinucleotide. To regenerate the consumed oxidized nicotinamide adenine dinucleotide, ethanol is produced. Moreover, the amino acids, except proline, may be used in grape juice fermentation to synthesize proteins. These amino acids are related to the final aromatic composition of wine. 

The high content of ethanol increases the proportion of sterols and unsaturated fatty acids [13]. Higher pH values concur with the catabolism of lactic acid and, therefore, a decrease in the wine acidity [6]. The reason for this phenomenon is that lactic acid is one of the producers of acetic acid (Figure 1 right (b)). This acetic acid is the main cause of sour wine. When the amount of this acetic acid increases, a sour flavor and vinegary odor are sensed. Table 1 presents the relationship between acetic acid and wine acidity; this information can be used to detect sour wine and to produce high-quality wines. The volatile acidity is detectable at 0.6–0.9 g/L and research shows that acidities higher than 1.2–1.3 g/L can result in an unpleasant taste [14,15,16,17].

### 2.1. TGS2620 Gas Sensor Module for Wine Analysis

Gas sensors based on metal oxide semiconductors (MOS) have been used in odor measurements to a greater extent than other sensors [18,19,20]. The process of gas detection with an MOS sensor involves the target gas inducing an electronic change on the oxide surface, which alters the sensor’s electrical resistance [21].

The sensor used in our work is the TGS2620. This MOS gas sensor was obtained from Figaro, the world leader in the gas-sensing innovation industry. The sensing element in TGS2620 consists of a metal oxide semiconductor layer formed on an aluminum oxide substrate along with an integrated heater. A simple electrical circuit can convert the conductivity change of the sensing element into an output signal corresponding to the gas concentration (Figure 2 (top)). Specifications for the TGS2620 are presented in Table 2.

The circuit diagram of the gas sensor module is shown schematically at the bottom of Figure 2. It consists of a conditioner circuit, powered by VSENS, followed by a buffer stage to adapt the impedance with an analog-to-digital converter. This buffer stage was carried out with the operational amplifier OP400 obtained from Analog Devices, which is a suitable choice for applications requiring low power consumption (less than 725 µA) and input bias current (less than 3 nA). The OP400 has a low input offset voltage (less than 150 µV) and a good noise voltage density (11 nV/√Hz at 1 kHz)

It is recommended to connect the sensor 24 h before the test to obtain a stable temperature response. During these tests, we will continue to power the gas sensor with a constant 5 V voltage. 

### 2.2. Data Acquisition Module

The card used for data acquisition is an NI-6221 card, with a maximum sample rate of 250 kS/s and 16 bits ADC resolution. The NI-6221 offers analog I/O, correlated digital I/O, two 32-bit counters/timers, and digital triggering. Following the pin planner, the output of the sensors was connected to the analogue port and the GND by means of port 67. The included NI-6221 driver and configuration utility simplify configuration and measurements.

The software used for the measurements was LabVIEW, which is a type of system-engineering software for applications that require testing, measurements, and control with rapid access to hardware and data insights. The implemented Virtual Instrument (VI) file format obtains the acquisition card voltage values in real time, creates a waveform graph, and outputs the data to a file for further analysis and interpretation. The DAQ Assistance module was used to manage the software system to obtain data with different configurations. Some parameters can be configured during the data acquisition process, including the rate, number of samples, and timeout.

### 2.3. Venturi Tool

High content of ethanol can saturate the sensors, which causes measurement errors in the wine samples. We designed a device that can mix the substance from the wine sample with air using the Venturi effect [22]. This can be explained by the Bernoulli equation:(1)$$\frac{V^{2}\rho }{2}+P+\rho gz=cte$$
where *P* is the static pressure of the fluid at the cross section, *ρ* is the density of the flowing fluid, *V* is the velocity of the fluid at the cross section, *g* is the acceleration caused by gravity, and *z* is the elevation head of the center of the cross section with respect to a reference level.

A 3D printer with PLA material was used to manufacture the device. The device is connected to the vessel, where the sample of acetic acid and wine is found. Volatile gas from sample and room-temperature air is diverted to a chamber, where both substances are mixed. This gas mixture reaches the sensor.

Figure 3 shows the TGS2620 gas sensor module, where we can see the four Figaro gas sensor mounting sockets and only a single gas sensor connected. This setting was used in tests with the Venturi tool, but three more sensors could be used.

## 3. Experimental Measurements

Acetic acid is the primary substance responsible for wine acidity and, therefore, it is necessary to carry out experiments with samples containing a certain known amount of acetic acid. For this reason, acetic acid mixtures with several concentrations were prepared in the laboratory.

For our experiments, we selected the percentage of the samples based on reference [14]. We prepared four samples with the following composition: wine (10 mL), wine (10 mL) + acetic acid (0.1 mL), wine (10 mL) + acetic acid (0.2 mL), and wine (10 mL) + acetic acid (0.4) mL. The samples were placed in little vessels, which were covered by a hermetic tap (see Figure 3d).

The gas sensor measurement procedure was divided into the exposure phase and cleaning phase. In the exposure phase, the sensor was inserted into the vessel where the wine and acetic acid were held. Subsequently, all other volatile compounds were removed in the cleaning phase to obtain a new correct measurement.

### 3.1. TGS2620 Gas Sensor Module Measurements

Experiments with the four samples were carried out in the following four ways: with and without the Venturi tool, and with the Venturi tool but with different airflows. 

To verify whether the use of the Venturi tool improves the measurement method, several experiments were carried out. Figure 4 (top) shows the considerable difference between their slopes. The information provided by the slopes is more useful than the difference between the initial and the final values. The sensor is almost always saturated in the measurement time before obtaining the final value.

Likewise, measurements were made using the Venturi tool with changes in the airflow. A small air injection pump was used to analyze this phenomenon. This pump inhales air from the room through an inlet and discharges the air to the outlet, which is directly connected to the Venturi tool’s air inlet. It can be seen that the output decreases when air is injected, as shown in Figure 4 (bottom).

### 3.2. Slope Analysis

From the results, it can be verified that the TGS2620 gas sensor module is sensitive to acetic acid changes. Figure 5 at the top shows results where samples of wine alone and wine mixed with acetic acid were used. We can clearly see that the slopes are different. Likewise, it can be verified that the amount of acetic acid in the samples affects the shape of the slope. Measurements with 0.1 mL and 0.2 mL acetic acid provide similar slope results; however, if the amount of acid is doubled (0.4 mL), the slope differs by a meaningful value (Figure 5 bottom).

An extensive measurement process was carried out to see if it was possible to distinguish between different samples. Both upward and downward slopes (see Figure 6) were analyzed exhaustively. From the results, it can be seen that simple classification algorithms can be used to distinguish between different wine samples, meaning that in practice we could determine the amount of acid in the wine.

The aim of our work, in addition to designing and manufacturing the device and therefore carrying out the experiments with the four samples, is to create a database of red wine sample features, including upward slopes and downward slopes. With the four different concentrations (wine, wine +1% acetic acid, wine +2% acetic acid, and wine +4% acetic acid), 84 tests were carried out for the database and the slopes were calculated using MatLab. We programmed MatLab code to calculate and compare the different slopes. In Figure 7, the result of the database with the mean values of the upward slopes can be seen. In addition, the minimum and maximum values, the median, and the first and third quartiles have been calculated. We can verify that the quantity of acetic acid in the wine determines the slope by comparing 0.1 and 0.2 mL of acetic acid, where the values are similar; however, if we pay attention to the 0.4 mL values, there is a larger difference.

## 4. Conclusions

In this work, a TGS2620 gas sensor module has been developed to identify the amount of acetic acid in wine over time. The measurement method has been improved by developing a tool that uses the Venturi effect. This tool is used to reduce the saturation of the sensor and to discriminate the different slopes. Further in-depth studies of the kinetics of the sensor are necessary.

In the last figure, it is demonstrated that there is a direct relationship between the slope of the sensor response and the acetic acid concentration. From the results, it can be seen that the upward slope increases as the acetic acid concentration increases in the samples. This provides a range of values where it is probable that ascetic acid values are similar.

In addition, it has been proven that ethanol converts to acetic acid in a short time period; when acetic acid is mixed with red wine and it turns sour, the samples that remain for several days have an increased concentration of acetic acid.

The aim was not to create a device that accurately measures the concentration of acetic acid in wine, but to create an inexpensive device that rapidly tests (between 2 and 5 min) to control the evolution of acetic acid concentration in red wine during its production process. It has been demonstrated that the TGS2620 gas sensor module is capable of differentiating several concentrations of acetic acid in wine, by analyzing upward and downward slopes, thereby providing useful information.

In future work, using a database with measurements at different time intervals during the production process of a certain variety of red wine, we aim to control the concentration of acetic acid. This would improve the repeatability of the production process of this variety of red wine.

## Figures and Tables

**Figure 1 sensors-23-02878-f001:**
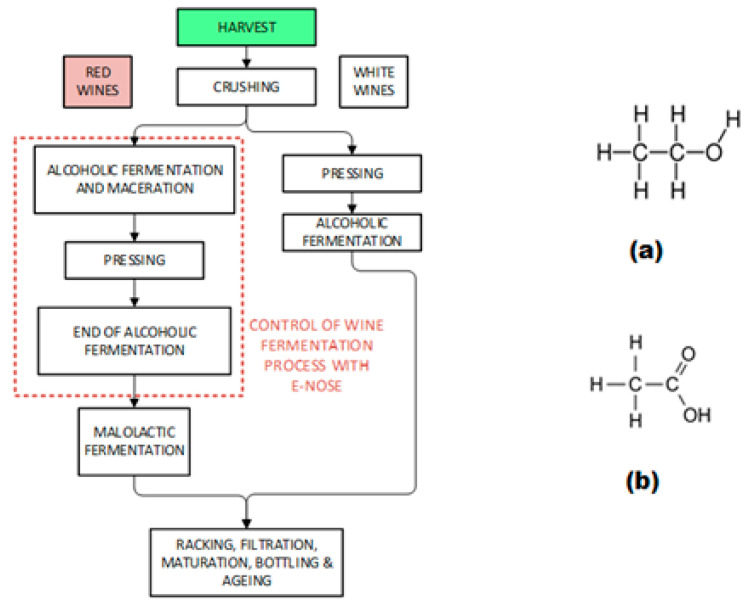
Flow chart of the wine process (**left** figure). Chemical structure of (**a**) ethanol and (**b**) acetic acid (**right** figure).

**Figure 2 sensors-23-02878-f002:**
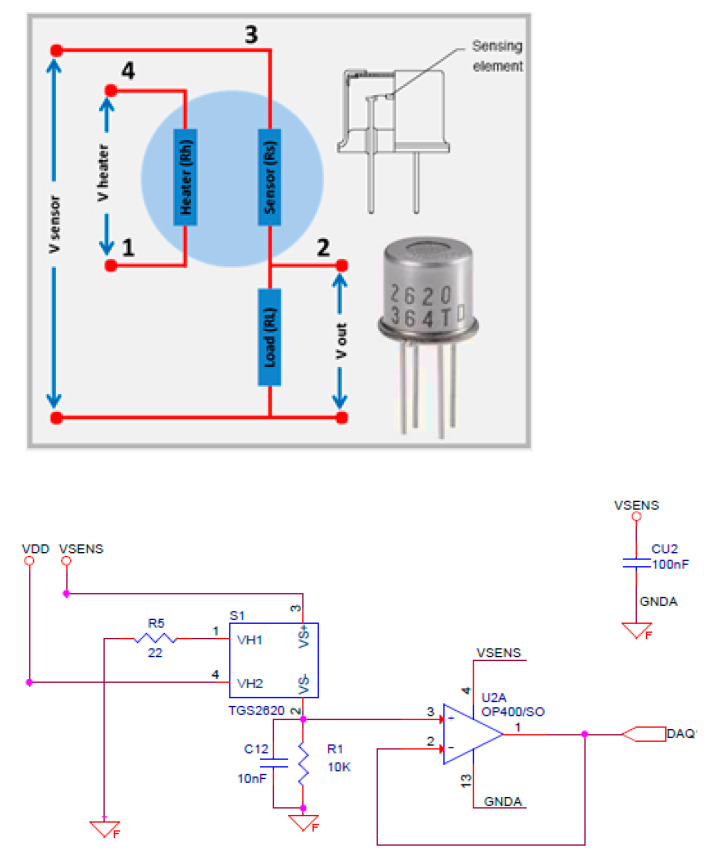
Circuit diagram for TGS2620.

**Figure 3 sensors-23-02878-f003:**
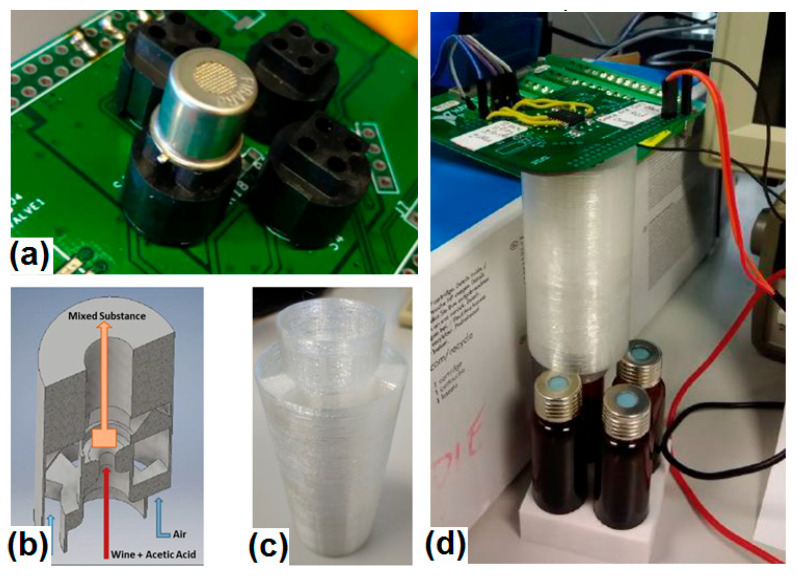
(**a**) The TGS2620 module; (**b**) the Venturi tool design; (**c**) the manufactured Venturi tool; (**d**) the measurement setup with the Venturi tool.

**Figure 4 sensors-23-02878-f004:**
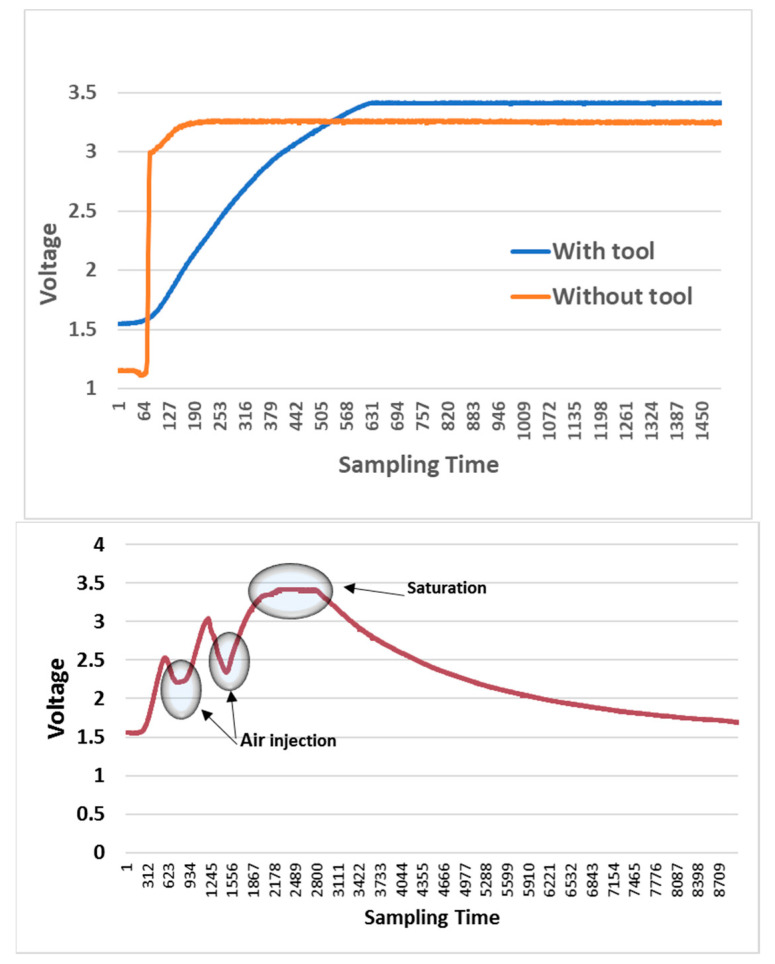
Measurements with and without Venturi tool (**top**). Airflow test using an injection pump (**bottom**). The sampling time is measured in tenths of a seconds.

**Figure 5 sensors-23-02878-f005:**
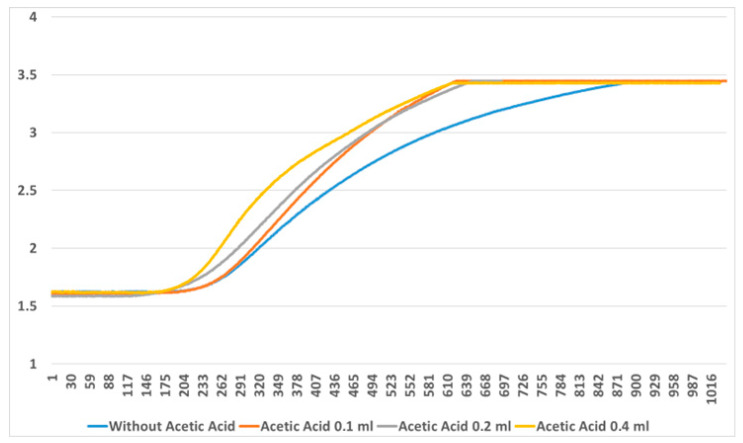
Wine without acetic acid and with 3 different acetic acid solutions (1, 2, and 4%). The sampling time is measured in tenths of a seconds.

**Figure 6 sensors-23-02878-f006:**
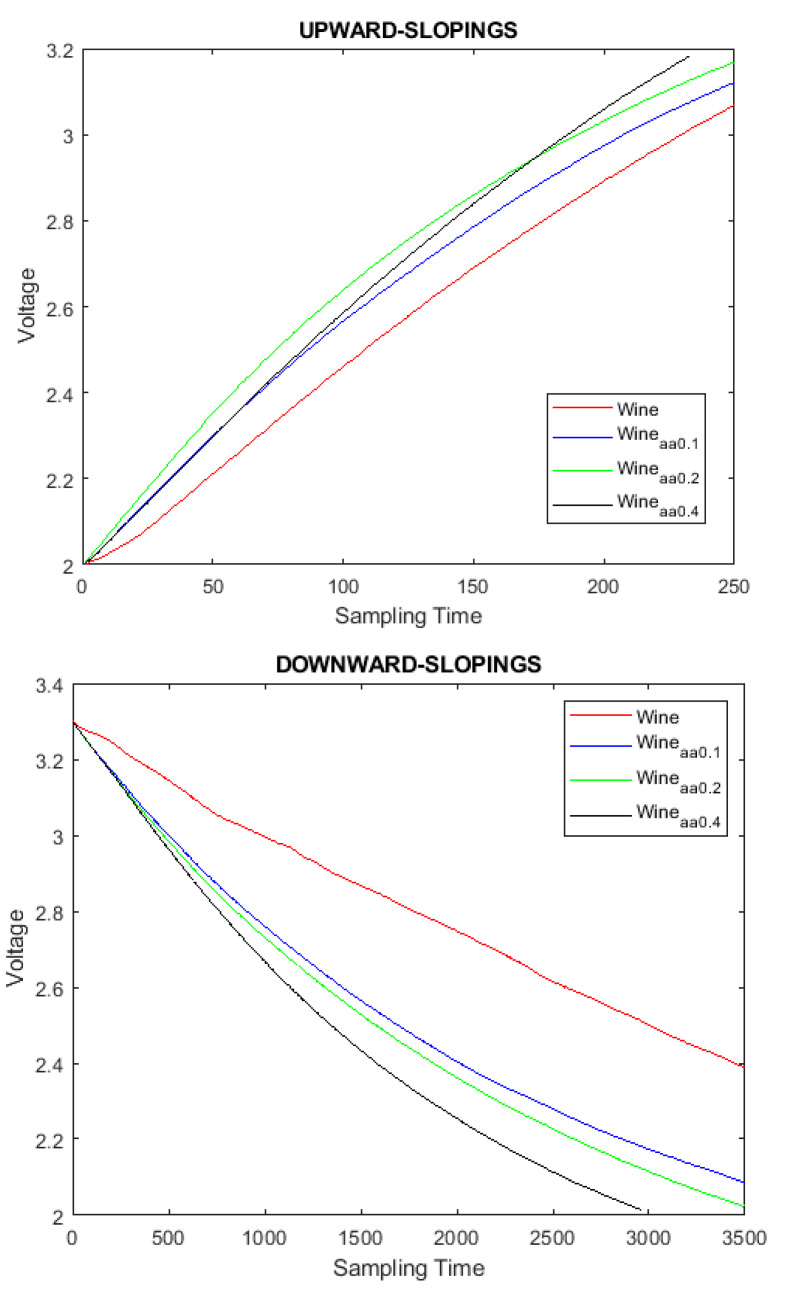
Details of upward slopes of wine with and without acetic acid (**top**); details of downward slopes of wine with and without acetic acid (**bottom**). The sampling time is measured in tenths of a seconds.

**Figure 7 sensors-23-02878-f007:**
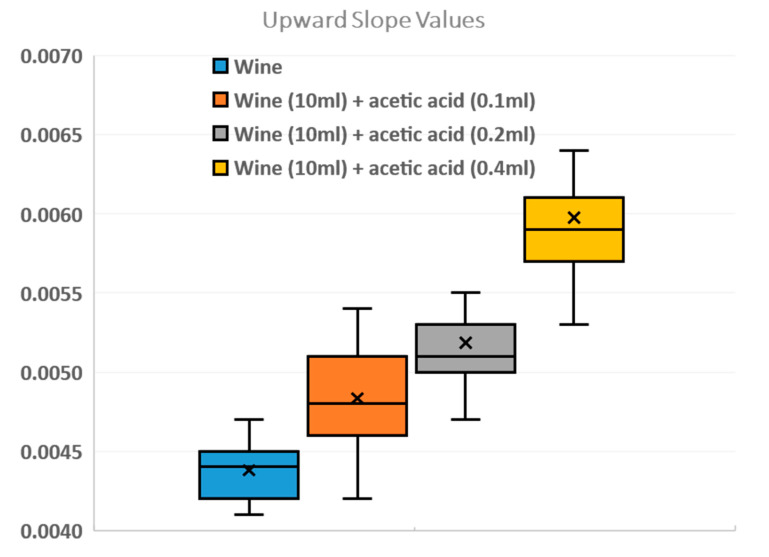
Box-and-whisker plot. Slope values for each test with mean and standard error, minimum, first quartile, median, third quartile, and maximum.

**Table 1 sensors-23-02878-t001:** Volatile acids and higher alcohols of wine [11].

Composition of Wine	Compound	Concentration (mg/L)	Descriptors
**Volatile fatty acids**	Acetic acid	150–900	Vinegar
	Hexanoic acid	Traces–37	Bitter, musty, vinegar
	Decanoic acid	Traces–54	Musty, sour, phenolic
**Higher alcohols**	Propanol	9–68	Alcohol
	Isobutyl alcohol	9–28	Alcohol

**Table 2 sensors-23-02878-t002:** TGS2620 specifications [20].

Model Number	TGS 2620
Target gases	Alcohol, solvent vapors
Detection range	50~5000 ppm
Heater voltage (VH) and Circuit voltage (VC)	5 V ± 0.2 V DC/AC
Load resistance (RL)	>0.45 kΩ
Heater resistance (RH)	83 Ω
Heater current	42 ± 4 mA
Heater power consumption (PH)	210 mW
Sensor resistance (Rs)	1~5 kΩ
Sensitivity	0.3~0.5

## Data Availability

Not applicable.

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
