# Peer review of "Evaluation of Red Wine Acidification Using an E-Nose System with Venturi Tool Sampling"

_sensors, 2023, doi:10.3390/s23062878_

Round 1

Reviewer 1 Report (Previous Reviewer 1)

In my opinion, the results presented are still preliminary. Appropriate comments and justifications must be included as indicated below:

Considering the author's response, I understand that the proposed approach is valid to assess the acidification of a wine sample over time (aging or degradation) or during its production process, but not to measure the concentration of acetic acid in any wine. The work approach must be clearly presented in the manuscript.

A title such as “Evaluation of red wine acidification by an e-nose system with Venturi-tool sampling” would be more appropriate (It is just a suggestion).

In Fig. 4-top there are two red curves. Which one corresponds to “with Venturi-tool” and which one corresponds to “without Venturi-tool”?

In Fig. 5-bottom it is better to include the curve “without acetic acid”, so taht Fig. 5-top can be excluded.

Fig. 6: Why did the authors not include the “wine-0.4ml” curve?

Fig. 7 is good: the authors demonstrated that there is a direct relationship between the slope of the sensor response and the acetic acid concentration. This result should be highlighted in the Abstract and Conclusions. However, two important pieces of information are missing: (1) How many tests were done for each concentration of acetic acid? and (2) How were the slopes of the curves obtained?

Line 191: please correct “red wine”.

In the Conclusions, the authors should include some “next steps” for the work.

Author Response

Reviewer 1

First of all, authors would like to thank the reviewer for his/her valuable comments and careful reading of the manuscript, which have been very useful for improving it in this new version.

In my opinion, the results presented are still preliminary. Appropriate comments and justifications must be included as indicated below:

Considering the author's response, I understand that the proposed approach is valid to assess the acidification of a wine sample over time (aging or degradation) or during its production process, but not to measure the concentration of acetic acid in any wine. The work approach must be clearly presented in the manuscript.

Thank you for the comment, we followed your suggestion, both in the abstract and in the conclusions,  in the new version of the manuscript.

A title such as “Evaluation of red wine acidification by an e-nose system with Venturi-tool sampling” would be more appropriate (It is just a suggestion).

Thank you for the comment, we followed your suggestion in the new version of the manuscript with the new title ‘Evaluation of red wine acidification by an e-nose system with Venturi-tool sampling’.

In Fig. 4-top there are two red curves. Which one corresponds to “with Venturi-tool” and which one corresponds to “without Venturi-tool”?

Thanks for the comment, we add the text to identifier the curves  in the figure 4.

In Fig. 5-bottom it is better to include the curve “without acetic acid”, so taht Fig. 5-top can be excluded.

Thanks for the comment, we include all the curves in one figure and remove the top.

Fig. 6: Why did the authors not include the “wine-0.4ml” curve?

Thanks for the comment, we include  the “wine-0.4ml” curve in Figure 6.

Fig. 7 is good: the authors demonstrated that there is a direct relationship between the slope of the sensor response and the acetic acid concentration. This result should be highlighted in the Abstract and Conclusions. However, two important pieces of information are missing: (1) How many tests were done for each concentration of acetic acid? and (2) How were the slopes of the curves obtained?

Thanks for the comment, we add the text to explain it. Lines from 209 to 215 in the new version.

Line 191: please correct “red wine”.

Thanks for the comment, we corrected it.

In the Conclusions, the authors should include some “next steps” for the work.

Thanks for the comment, we add the text about future work, lines from 240 to 243.

Reviewer 2 Report (Previous Reviewer 3)

The manuscript is better, but full of naive mistakes. It is necessary to revise English throughout the manuscript.

Author Response

Reviewer 2

First of all, authors would like to thank the reviewer for his/her valuable comments and careful reading of the manuscript, which have been very useful for improving it in this new version.

Comments and Suggestions for Authors

The manuscript is better, but full of naive mistakes. It is necessary to revise English throughout the manuscript.

We appreciate the reviewer valuable suggestion. We recheck the paper and rewrite some sentences to correct it, We hope that the writing errors that the reviewer commented have been corrected.

This manuscript is a resubmission of an earlier submission. The following is a list of the peer review reports and author responses from that submission.

Round 1

Reviewer 1 Report

The manuscript presents a system that aims to measure the amount of acetic acid in red wine, analyzing its volatile gases.

The introduction and contextualization are good and the proposal to use the Venturi effect to get the gas sample for analysis is a good insight.

However the work presents a serious methodological problem: it is not possible to quantify a single gas species (acetic acid) using only one MOX type gas sensor (TGS2620), as this kind of sensor has a poor selectivity, already known and studied for decades. Wine releases many volatile species, as the authors mentioned. So the sensor response is influenced by all species.

The results presented are still preliminary. The proposed system needs to be improved with an array of different sensors and adequate data processing to be presented as wine acetic acid sensor. Therefore, I consider that the manuscript is not suitable for publication in the Sensors journal in its present form.

Reviewer 2 Report

This manuscript “Enhancing acetic acid measurements with Venturi-tool to characterize red wine quality” by Esmeralda Hernández et al. reported a sensor using TGS2620 module to distinguish between different grades of acetic acid in red wine. The whole logic of the manuscript seems a little bit chaos, the experimental data is insufficiency. There are some comments for authors to improve their manuscript.

1. Parameters in Fig. 2 are not clear enough for readers. Besides, what is the output voltage range of TGS2620? Why choose voltage follower circuit instead of an amplifier, although they select an operational amplifiers OP400?

2. Line 122 mentioned a e-Nose named MOOSY32, but the Ref [22] webpage shows “Not found, error 404”, please update it or select another comparison.

3. Line 124-125 said that “Figures 3 and 4 show the captured signals, truncated to two decimal places, that were used to identify patterns.”, however, Fig. 3 is some pictures of TGS2620.

4. Line 128 mentioned that “we propose substituting the ADC National Instruments boards with a 128 sigma-delta ADC implemented on an FPGA,” and Line 130 “target is to obtain an ADC with a minimum of 9 bits of resolution”, but Sec. 2.2, a NI-6221 with 16 bits ADC resolution was selected. Please double check it!

5. Fig. 4 shows the output voltages which can reach 3.5V, why Line 127 said that “they are always positive, lower than 2 V and with some offset”?

6. What is the unit of sampling time in Fig. 4, 5 and 8? Second or minutes?

7. What is the stability and detection limit of the sensors? Comparing with the MOOSY32, is there any advantages?

Reviewer 3 Report

I think that the quality of the Manuscript ID: sensors-2124889 entitled "Enhancing acetic acid measurements with Venturi-tool to characterize red wine quality" is good and can be accepted with some major revisions:

The manuscript is full of naive mistakes. Some examples:

Line 9: at consumption time should be the consumption;

Line 13-14: in the wine should be in wine;

Lines 26-36: the paragraph is difficult to read. Please rewrite using a suitable English. Moreover, it is not clear to me why the authors cite the reference number 3 which is related to Khasi Mandarin orange. Please, specify;

Line 74: there red box indicate phase where a wine tester is involved. should be where red box indicates the phase which involves a wine tester;

Line 98: Table 1 present should be Table 1 presents;

Line 115: Specifications of the TGS2620 presents in Table 2 should be Specifications of the TGS2620 are presented in Table 2;

Line 125: As can be seen, should be As it can be see;

Line 146: The card used for data acquisition, is a NI-6221 card. The data acquisition card has should be The card used for data acquisition is a NI-6221 card which has;

Lines 177-178: The sentence is not clear, please rewrite;

Line 182: with well determined: this expression is not clear, please specify;

Lines 184-187: the paragraph is hard to follow. Please specify. The last sentence (lines 185-187) lacks the principal verb;

Lines 196-197: The Venturi-tool has been tested by changing various parameters in the measures. One of these parameters is airflow. should be The Venturi-tool has been tested by changing various parameters in the measures, including the airflow;

Lines 199-200: It can be seen while injecting air how the voltage decreases, as shown in Figure 4 (bottom). should be It can be seen that when the air is injected air the voltage decreases, as shown in Figure 4 (bottom).;

Lines 213-214: Comparison of 0.1 and 0.2 ml amounts of acetic acid, the result is similar should be Comparing 0.1 and 0.2 ml amounts of acetic acid, it can be see that result is similar;

Lines 221-222: The sentence is not clear. Please, rewrite;

Lines 232-234: the sentence is not clear, please rewrite;

Line 235: cause should be causes;

Line 236: turn should be turns

And so on

It is necessary to revise English throughout the manuscript.